# Right Vertebral Artery Intermittent Flow Reversal Due to Innominate Artery Dissection

**DOI:** 10.3390/diagnostics15131668

**Published:** 2025-06-30

**Authors:** Corrado Tagliati, Alessia Quaranta, Marco Fogante, Stefania Lamja, Alfonso Alberto Matarrese, Davide Battista, Giulio Cocco, Giuseppe Lanni, Alberto Rebonato, Fabiola Principi, Giulio Argalia, Antonio Corvino, Iacopo Carbone, Ernesto Di Cesare, Nicolò Schicchi

**Affiliations:** 1AST Ancona, Ospedale di Comunità Maria Montessori di Chiaravalle, Via Fratelli Rosselli 176, 60033 Chiaravalle, Italy; 2AST Macerata, Cardiologia, Distretto Sanitario di Civitanova Marche, Via Abruzzo, 62012 Civitanova Marche, Italy; alessiaquaranta84@gmail.com; 3Maternal-Child, Senological, Cardiological Radiology and Outpatient Ultrasound, Department of Radiological Sciences, University Hospital of Marche, Via Conca 71, 60126 Ancona, Italy; marco.fogante89@gmail.com (M.F.); giulio.argalia@gmail.com (G.A.); nicolo.schicchi@ospedaliriuniti.marche.it (N.S.); 4Department of Biotechnological and Applied Clinical Sciences, University of L’Aquila, Via Vetoio, 67100 L’Aquila, Italy; stefanialamja@gmail.com (S.L.); ernesto.dicesare@univaq.it (E.D.C.); 5AST Ascoli Piceno, Cardiologia, Ospedale Mazzoni, Via degli Iris 1, 63100 Ascoli Piceno, Italy; alfonsomatarrese@gmail.com; 6Department of Services, UOSD Radiology, San Liberatore Hospital, Viale Risorgimento, 64032 Atri, Italy; davide.battista@aslteramo.it; 7Department of Neuroscience, Imaging and Clinical Sciences, University “G. d’Annunzio”, 66100 Chieti, Italy; cocco.giulio@gmail.com; 8Department of Services, UOSD Diagnostica per Immagini Teramo, Ospedale Civile Giuseppe Mazzini, Piazza Italia, 64100 Teramo, Italy; giuseppe.lanni@aslteramo.it; 9AST Pesaro-Urbino, Radiologia, Ospedale San Salvatore, Piazzale Cinnelli 1, 61121 Pesaro, Italy; alberto.rebonato@sanita.marche.it; 10AST Ancona, Radiologia, Ospedale Santa Casa di Loreto, via San Francesco 1, 60025 Loreto, Italy; fabiola.principi@sanita.marche.it; 11Medical, Movement and Wellbeing Sciences Department, University of Naples “Parthenope”, 80133 Naples, Italy; an.cor@hotmail.it; 12Department of Radiological, Oncological and Pathological Sciences, Academic Diagnostic Imaging Division, I.C.O.T. Hospital, Sapienza University of Rome, Via F. Faggiana 1668, 04100 Latina, Italy; iacopo.carbone@uniroma1.it

**Keywords:** vertebral artery, flow reversal, intermittent, sporadic, irregular, dissection, flap, innominate artery

## Abstract

Here, we describe a case of an asymptomatic 73-year-old female patient who suffered from type A acute aortic dissection with epiaortic arteries involvement and underwent surgical operation 9 years ago. A follow-up color Doppler ultrasound revealed a right vertebral artery intermittent flow reversal due to innominate artery dissection. To our knowledge, no previous studies have reported this intermittent flow reversal; therefore, supra-aortic trunks should be considered among the possible causes of vertebral artery flow reversal.

**Figure 1 diagnostics-15-01668-f001:**
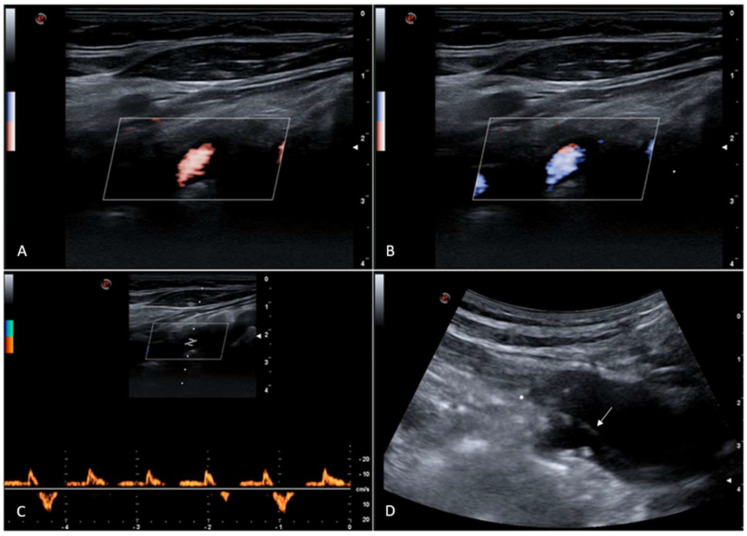
Color Doppler ultrasound shows antegrade (**A**) and retrograde (**B**) right vertebral artery flow; Doppler spectral waveform shows intermittent and irregular right vertebral artery flow reversal (**C**). Innominate artery dissection can be demonstrated ((**D**), Appendix A), with a dissection flap just near the right subclavian artery origin. It can be supposed that dissection flap irregular movements can cause a sporadic reduction of end-systolic subclavian artery pressure, which could be at the base of the random right vertebral artery flow reversal occurrence. The 73-year-old female patient was asymptomatic at the time of the ultrasound examination; she suffered from spontaneous type A acute aortic dissection with epiaortic arteries involvement and underwent surgical operation 9 years ago. Type A aortic dissection is a life-threatening surgical emergency with a suggested incidence of about 5 per 100,000 person-years, sometimes associated with supra-aortic trunk involvement [1,2,3,4]. Long-term outcomes of this disease are improving over time, and it is known that sometimes false channels remain patent after surgery [5,6]. A previous study reported an aortic dissection extending to the innominate and right common carotid arteries which showed blood passing from the false lumen of the distal right common carotid artery into the true lumen with antegrade flow in the false lumen but reverse flow in the true channel, the latter supplying the subclavian artery [7]. To our knowledge, no previous studies have reported this intermittent flow reversal; therefore, supra-aortic trunks dissection should be considered among the possible causes of vertebral artery flow reversal.

## Data Availability

Data are contained within the article.

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
