# Peer review of "Right Vertebral Artery Intermittent Flow Reversal Due to Innominate Artery Dissection"

_diagnostics, 2025, doi:10.3390/diagnostics15131668_

Round 1
Reviewer 1 Report
Comments and Suggestions for Authors
My comments mainly relate to the language writing as it needs to be polished as there are many grammatical errors. Just list a few:
Here, we describe the case of an asymptomatic 73-year-old female patient who suffered 34 her type A acute aortic dissection with epiaortic arteries involvement and surgical operation 9 years before. A follow-up color-Doppler ultrasound revealed a right vertebral artery intermittent flow reversal due to innominate artery dissection. To our knowledge, no previous articles reported this intermittent flow reversal; therefore, supra-aortic trunks should be considered among the possible causes of vertebral artery flow reversal.
We describe a case of…. who suffered from type A acute aortic dissection. 9 years before-9 years ago. To our knowledge, no previous studies have reported…
Language editing is suggested.
Comments on the Quality of English LanguageLanguage editing is needed to improve the readability.
Author Response
Dear Reviewer,
Thank you very much for taking the time to review this manuscript. Please find the responses below and the corresponding corrections highlighted in the resubmitted file.
My comments mainly relate to the language writing as it needs to be polished as there are many grammatical errors.
R: Thank you very much for your suggestion.
Here, we describe the case of an asymptomatic 73-year-old female patient who suffered 34 her type A acute aortic dissection with epiaortic arteries involvement and surgical operation 9 years before. A follow-up color-Doppler ultrasound revealed a right vertebral artery intermittent flow reversal due to innominate artery dissection. To our knowledge, no previous articles reported this intermittent flow reversal; therefore, supra-aortic trunks should be considered among the possible causes of vertebral artery flow reversal.
We describe a case of…. who suffered from type A acute aortic dissection. 9 years before-9 years ago. To our knowledge, no previous studies have reported…
Language editing is suggested.
R: Thank you very much for your suggestion. Grammatical errors were modified.
Reviewer 2 Report
Comments and Suggestions for Authors
This case report is quite interesting, with representative and insightful images along with clear and concise legends. When available, CTA or DSA imaging illustrating the innominate artery dissection would provide a more detailed assessment of the lesion.
Author Response
Dear Reviewer,
Thank you very much for taking the time to review this manuscript. Please find the response below and the corresponding changes highlighted in the resubmitted files.
This case report is quite interesting, with representative and insightful images along with clear and concise legends. When available, CTA or DSA imaging illustrating the innominate artery dissection would provide a more detailed assessment of the lesion.
R: Thank you for your suggestion. As the CT image is not so nice and a little bit blurred, we added it in the manuscript as supplementary material.
Round 2
Reviewer 1 Report
Comments and Suggestions for Authors
Thank you for addressing my comments in the revision and it is acceptable for publication.